# The measurement of self-regulation in the Adolescent Brain Cognitive Development (ABCD) Study

Merle Johanna Marek[1]*, Axel Heep[2,3], Andrea Hildebrandt[1,3]*

1 Psychological Methods and Statistics, Department of Psychology, School of Medicine and Health Science, Carl von Ossietzky Universität Oldenburg, Oldenburg, Germany, 2 Paediatrics, Department of Human Medicine, School of Medicine and Health Sciences, Carl von Ossietzky Universität Oldenburg, Oldenburg, Germany, 3 Research Center Neurosensory Science, Carl von Ossietzky Universität Oldenburg, Oldenburg, Germany

* merle.marek@uol.de (MJM); andrea.hildebrandt@uol.de (AH)

## Abstract

To facilitate future research on self-regulation and related brain-behavior associations, we aimed to establish a psychometric model of self-regulation in the largest open neuroimaging dataset to date, the Adolescent Brain and Cognitive Development (ABCD; https://abcdstudy.org/). Given the measures adopted in the ABCD study, we tested three theoretically defensible and applicable psychometric models of self-regulation. The dual-process theory provided the framework for postulating the models to be tested. This theory states that successful self-regulation occurs in case of a balanced state between bottom-up 'hot' and top-down 'cool' processes in favor of achieving goals. Based on the results, we recommend a measurement model with three correlated first-order factors: Hot, Cool and Executive Functions. The model successfully predicted academic achievement both at the time of self-regulation assessment and two years later, and its robustness across smaller samples was confirmed. Given its factorial and predictive validity, we recommend the adoption of the established model for future research on self-regulation and its neural correlates based on the ABCD dataset. Given the measures adopted in the ABCD study, a theoretically desirable bifactor model with a general self-regulation factor and nested Hot and Cool factors cannot be reliably established.

## Introduction

Based on the advances in large-scale neuroimaging initiatives on representative samples, such as the Human Connectome Project (HCP; http://www.humanconnectomeproject.org), the Alzheimer's Disease Neuroimaging Initiative (ADNI; https://adni.loni.usc.edu), or the Adolescent Brain Cognitive Development (ABCD; https://abcd-study.org) study, multivariate analyses of brain-behavior associations are promoted.

**Data availability statement:** Due to data access restrictions outlined by the ABCD study, the authors are unable to publicly share the study's dataset. However, the ABCD study embraces an open science approach and data access can be requested by any eligible researcher with a valid research use of the data (https://nda.nih.gov/nda/access-data-info). Please note that the full analysis code can be accessed at https://osf.io/nzjf7/.

**Funding:** Data used in the preparation of this article were obtained from the Adolescent Brain Cognitive Development (ABCD) Study (https://abcdstudy.org), held in the NIMH Data Archive (NDA). This is a multisite, longitudinal study designed to recruit more than 10,000 children age 9-10 and follow them over 10 years into early adulthood. The ABCD Study is supported by the National Institutes of Health and additional federal partners under award numbers U01DA041048, U01DA050989, U01DA051016, U01DA041022, U01DA051018, U01DA051037, U01DA050987, U01DA041174, U01DA041106, U01DA041117, U01DA041028, U01DA041134, U01DA050988, U01DA051039, U01DA041156, U01DA041025, U01DA041120, U01DA051038, U01DA041148, U01DA041093, U01DA041089, U24DA041123, U24DA041147. A full list of supporters is available at https://abcdstudy.org/federal-partners.html. A listing of participating sites and a complete listing of the study investigators can be found at https://abcdstudy.org/consortium_members/. ABCD consortium investigators designed and implemented the study and/or provided data but did not necessarily participate in the analysis or writing of this report. This manuscript reflects the views of the authors and may not reflect the opinions or views of the NIH or ABCD consortium investigators.

**Competing interests:** The authors have declared that no competing interests exist.

Understanding the link between brain systems and networks and human behavior is central to cognitive-behavioral neuroscience and offers critical insights into mechanisms underlying cognition, personality, and psychiatric conditions (e.g., [1–3]). To leverage the potential of these datasets for assessing brain-behavior associations, it is required to estimate latent traits of interest based on a priori fixed measurements included in the respective study. Estimating traits is the core challenge of psychometrics and is particularly daunting in cases of vaguely defined and broad psychological constructs, such as self-regulation (SR). Since researchers analyzing data from these neuroimaging initiatives often have expertise in neuroimaging but less so in psychometrics, proposing well-established and robust psychometric models—along with implementation code—can enhance the statistical analysis of brain-behavior associations and support valid conclusions in neuropsychological research using the ABCD study. The aim of this study was to foster a well-grounded and consistent construct representation of SR in future brain-behavior association studies in the ABCD dataset – today's largest open longitudinal neuroimaging study. To reach this goal, we investigated alternative measurement models and provided code that can be used in future analyses of SR's neural correlates. Furthermore, we emphasize that the derived theoretically plausible measurement model might be applicable to different multivariate datasets within a similar study design.

### The construct of self-regulation

SR refers to the ability to actively guide oneself toward a desired emotional, behavioral, or cognitive state (e.g., [4]). It involves adapting internal states (i.e., emotions and cognitions) and behaviors to align with contextual demands [5]. Context, in this regard, encompasses situational factors, social expectations, and personal goals. SR is a complex, multidimensional construct that depends on dynamic interactions between traits (e.g., temperament, impulsivity) and states (e.g., hormonal and physiological conditions) and involves various subskills, such as delay of gratification, executive functions (EFs) and self-control (see [5] for a glossary of related terminology). Although SR and self-control are sometimes used interchangeably, they are more accurately understood as distinct but related concepts. Self-control specifically addresses behavioral regulation in the face of conflicts between competing goals, such as resisting short-term rewards to prioritize long-term outcomes. In contrast, SR encompasses a broader set of processes, including planning, goal-setting, monitoring, in addition to the execution of behavior [4,6,7]. Unlike self-control, SR is not necessarily dependent on the presence of a conflict between competing goals [4]. Given this complexity, SR has been challenging to define consistently, often falling prey to jingle-jangle fallacies, where similar terms are used interchangeably or distinct concepts are conflated (e.g., [8]). Despite definitional challenges, SR has drawn significant interest across biological, differential, developmental, and clinical psychology due to its malleability [9] and its predictive power for life outcomes such as academic achievement, social relationship quality, and substance-use behavior [10,11].

SR is commonly conceptualized through the dual-process theory [4,12,13], in particular, when considering its neural underpinnings. According to this theory,

successful SR is achieved by balancing two domain-specific systems. The first one reflects a cognitive, rational, top-down, or so-called cool component that is mostly related to control processes focused on long-term goals. EFs form an integral part of this cool components, as they enable the cognitive control necessary for effective SR. However, EFs are not synonymous with SR; while critical, they are not sufficient for SR and can also support other psychological goals [5,14]. The factor structure of EF, detailed in [15], comprises three core components: cognitive flexibility or set shifting, which involves the ability to switch between tasks, mental sets or operations; working memory, which allows active updating, operating, and monitoring of mental representations; and inhibitory control, the deliberate suppression of prepotent responses. The second component of SR encompasses emotional, impulsive, bottom-up, or hot processes often oriented towards short-term rewards. Effective SR depends on the ability to appropriately balance and adapt these two components to the demands of the context.

The dual-process model of SR has been widely applied in studies investigating brain-behavior associations. The cool component has been related with prefrontal cortex activation (e.g., [16]), and the hot component was additionally found to be associated with activation in subcortical limbic structures, such as the ventral striatum and the amygdala (e.g., [17,18]). The strength of fronto-striatal connections between these regions has been linked to SR in older adults [19] and (pre-)adolescents [20–22]. Furthermore, an imbalanced trajectory describing the development of connections between striatal and prefrontal brain regions, resulting in increased reward responsivity and comparatively weaker control processes, has been suggested as the root of dysregulated behavior in adolescence [23]. This underlines the considerable importance of studying a dual-process model of SR in the field of neurobehavioral research as a prerequisite for future brain-behavior research in such large-scale datasets as the ABCD.

## The measurement of self-regulation

SR encompasses a multitude of subskills that collectively define the ability to regulate oneself. Efforts to establish a comprehensive nomological network model for SR have provided valuable insights (e.g., [24,25]), but the complexity of this task remains significant. Research on the convergent validity of SR measures reveals weak correlations between different content- and methodological-domains [24,26].

SR subskills are typically assessed using different methods: some are measured with questionnaires [27,28], others are assessed by means of behavioral tasks [29] or by combining both methods [24]. These approaches are complementary, and an exhaustive model should incorporate both. Heterogeneous indicators of SR from various content domains – which are often assessed using different measurement approaches – enhance content validity. However, the consistently low correlations between survey and behavioral SR indicators [24] point to issues with convergent validity.

Each method has distinct advantages and limitations. While survey-based indicators of SR typically demonstrate high reliability and are, therefore, recommended for the study of individual differences in association with outcomes relevant for daily life [30], behavioral tasks often exhibit low reliability [30,31]. However, behavioral measures of SR are less influenced by introspection ability or socially desirable responding. They also typically assess more isolated constructs (e.g., response inhibition), which often correspond to lower variability in brain correlates. Therefore, behavioral tasks may be more appropriate for studying brain-behavior relationships [32], which is one of the main objectives of the ABCD dataset. In light of these considerations, combining task-based and survey-based SR measures from different content domains poses both opportunities and risks. To address these, we used SR measures from the ABCD study to model their covariance structure using various latent variable models, comparing several defensible approaches to account for the advantages and limitations of each method.

## The measurement of self-regulation in the ABCD study

Conducting a large-scale study to apply extensive test batteries including behavioral and survey indicators of SR along with neuroimaging data acquisition is highly resource-intensive. The trade-off between resource-investment related to extensive

test batteries on the one hand, and content validity of the measured constructs on the other hand, is increasingly balanced by open large-scale neuroimaging initiatives as those mentioned above. To take full advantage of these datasets, psycho-metric models based on the measures used in these studies need to be established. The ABCD study includes neural data and a > 8,5 h test battery of > 11,000 nine- to ten-year-old pre-adolescents at a baseline assessment. With its variety of brain measures, bio-specimens, and psychological data, the ABCD dataset offers ideal conditions for a thorough investigation of complex constructs such as SR, as well as biological and environmental determinants of individual differences in SR. Establishing psychometric models for the constructs covered in the ABCD study will ideally encourage researchers to apply consistent models across multiple studies, thus promoting a cumulative understanding of SR in adolescence.

We evaluated all available instruments of the ABCD study related to mental health and neurocognition based on their theoretical overlap with the construct of SR or its subconstructs. Potentially related constructs were identified based on [5] and [33], which include a glossary of major terms and mental processes related to SR and an overview of task and survey-based indicators of SR, respectively. We identified three questionnaires measuring hot and cool facets of self-reported impulsivity and inhibition/reward seeking, as well as caregiver-reported attention problems, and three EF tasks as potential indicators of SR.

## The present study

The research question guiding this modeling endeavor asks whether SR (in the ABCD dataset) can be represented by two distinct domain-specific systems, i.e., one rational or cool component, and one emotionally-driven or hot component, as postulated by the dual-process theory. We conceptualized and tested several defensible measurement models of SR in the ABCD study within this theoretical framework. By doing so, we aim to facilitate cumulative knowledge-gain on SR based on future studies involving the ABCD data. We want to highlight that our goal was neither to test the dual-process theory of SR, nor to find the best possible model of SR or compare different theories. Rather, our aim was to test how well SR, as conceptualized by the dual-process theory, is measurable by the available indicators in the rich and influential ABCD dataset. At the same time, we aim to foster multivariate modeling in personality research by establishing constructs with heterogeneous observables and high factorial and predictive validity. Thus, we modeled SR in the ABCD dataset based on several indicators from different assessment modalities (i.e., self-reports, caregiver-report questionnaires, and behavioral tasks).

Based on the theoretical assumption that all selected measures indicate – at least to some extent – the general construct of SR, our initial model included a general factor that reflected potential common variance of all indicators in addition to the factors representing domain-specific traits. However, recognizing the challenges in achieving convergent validity across different measurement modalities – specifically self- and other-reports as compared with tasks – we progressively relaxed this assumption. In Model 2, we separated behavioral tasks from survey-based measures while retaining a general factor. Finally, we tested a model without a general SR factor, allowing for the possibility that SR might be better represented without a general construct. The three models that were tested are shown in Fig 1 and described in more detail here:

1) We estimated a bifactor model with a domain-general SR factor ($G_{SR}$) and three orthogonal nested factors. The first two nested factors comprise the hot and cool components of SR based on survey data and are named Hot and Cool, respectively. The third nested factor reflects EF measured by behavioral tasks and is referred to as EF. Even though the Cool and EF factors are conceptually closely related according to the dual-process theory, due to differing assessment domains in the ABCD study (tasks for EFs and surveys for other cool indicators), we expected their correlation to be too small to be combined within a common factor [24].

2) Given expectedly low correlations between survey- and task-based indicators of SR, the second defensible model is similar to the first one, except for the fact that the EF factor was assessed in a separate model. Thus, the bifactor

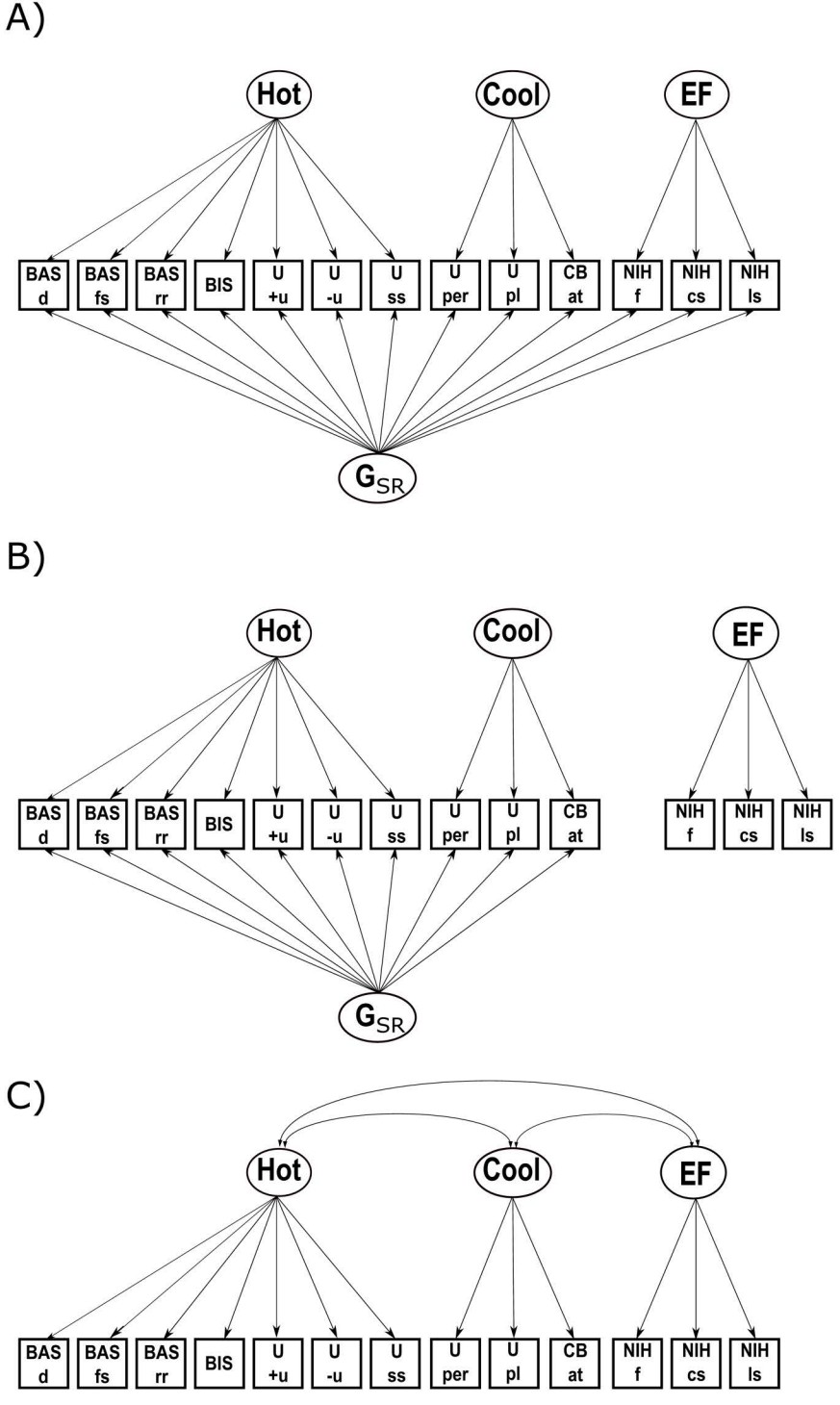

**Fig 1. Defensible measurement models of self-regulation that were compared with each other.** (A) Model 1: bifactor model with a domain-general SR factor ($G_{SR}$) and three orthogonal nested factors (B) Model 2: bifactor model with $G_{SR}$ and three orthogonal nested factors only including survey-based indicators, behavioral EF tasks are measured separately (C) Model 3: model with three correlated first order factors and no domain-general factor; Hot/Cool/EF: domain-specific factors for hot/cool/executive functions components of SR; U: UPPS-P; +u: positive urgency; -u: negative urgency; ss: sensation seeking; pl: planning; per: perseverance; BIS/BAS: Behavioral Inhibition and Behavioral Approach System Scale, inhibition; rr: reward responsivity; fs: fun seeking; d: drive; CB: Parent Child Behavior Check List; at: attention problems; NIH: NIH toolbox behavioral tasks; f: flanker task; cs: dimensional card sort task; ls: list sort working memory test.

model included survey-based $G_{SR}$, Hot and Cool performance. Behavioral EF tasks were measured separately. This model assumes SR performance based on surveys and behavioral tasks to be uncorrelated.

3) Even though hot and cool components of SR are thought to share common processes, there is conflicting evidence whether they can be combined in a general factor. While some studies found evidence for a common factor of hot and cool processes across behavioral and survey-based indicators [29], others found no commonality between hot and cool behavioral tasks [34]. Therefore, in the third defensible model, Hot, Cool, and EF were estimated as correlated first-order factors in a model without the domain-general factor $G_{SR}$.

To test the predictive validity of the model, we further expected that SR is associated with daily-life outcomes. In children and adolescents, academic achievement is greatly influenced by SR [35] and was, therefore, treated as the outcome of interest. Also, we expected that the measurement model of SR is robust across multiple smaller samples and independent from family-membership of the participants.

## Materials and methods

### Participants

All data was obtained from the Adolescent Brain Cognitive Development (ABCD) Dataset (https://abcdstudy.org), data release 4.0. The ABCD study is the largest longitudinal study on children's development to date, including 11,876 participants at the baseline assessment measured at 21 sites across the U.S. Baseline measurements started in 2016 and included participants aged nine to ten. Ethical approval of study protocols was given by the central Institutional Review Board, University of California, San Diego and individual study sites relied on local IRB approval. Caregivers provided written informed consent and children provided written assent [36]. In the present study, SR at the baseline assessment will be considered. Permission to access the anonymized data was first granted to the authors of this study on November, 22nd 2021 and no information that could identify individual participants were ever available to the authors.

The sampling was designed to mimic sociodemographic patterns of the U.S. population as closely as possible, while oversampling siblings and especially twins. Baseline physical characteristics of the participants are described in [36]. Exclusion criteria for the overall participation were children's lack of English language fluency, any condition preventing neuroimaging, a gestational age < 28 weeks or a birth weight <1200 g, hospitalization of more than a month after birth complications, and severe mental illnesses [36]. One individual was excluded from the present analysis due to missing data. In the whole data matrix, there were 2,836 (1.3%) missing values. At the baseline assessment, participants and their primary care takers were invited to the research site to undergo a test battery including questionnaires, behavioral and neurocognitive tasks, multimodal neuroimaging and bio-specimen sampling.

### Measurement instruments

All instruments included in the baseline assessment of the ABCD study that address a subcomponent of SR [5,33], were chosen as potential indicators of SR. Some instruments that were selected as indicators of SR were reverse-coded prior to the analysis to consistently indicate better SR at higher scores. First, the UPPS-P ((negative) urgency, premeditation, perseverance, sensation seeking – positive urgency) questionnaire [37,38] was selected, which captures impulsivity on five scales with four items each: negative urgency (e.g., '*When I feel bad, I often do things I later regret in order to make myself feel better now.*'), (lack of) premeditation (e.g., '*I like to stop and think about things before I do it.*'), (lack of) perseverance (e.g., '*I almost always finish projects that I start.*'), sensation seeking (e.g., '*I enjoy taking risks.*'), and positive urgency (e.g., '*I tend to act without thinking when I am very, very happy.*'). Next, we chose the Youth Behavioral Inhibition/ Behavioral Approach System Scale (BIS/BAS, [39–41]) with BAS scale scores for fun seeking (four items, e.g., '*I often do things for no other reason than they might be fun.*'), drive (four items, e.g., '*When I want something, I usually go all the way to get it.*'), reward responsiveness (five items, e.g., '*I get thrilled when good things happen to me.*'), as well as

inhibition (BIS, seven items, e.g., '*Even if something bad is about to happen to me, I rarely experience fear or nervousness.*'); and the attention scale score of the Child Behavior Checklist (CBCL; [42]; ten items, e.g., '*Can't concentrate, can't pay attention for long*'). Additionally, three behavioral tasks from the NIH Cognition Assessment Toolbox [43] were selected as indicators of EF. The Flanker task captures inhibitory cognitive control by presenting target and flanking arrow-stimuli that can be either congruent or incongruent regarding the direction they point at. Performance is scored based on speed and accuracy of identifying the direction of the target, independently of the flanking stimuli. The Dimensional Card Sort task captures set shifting and cognitive flexibility and presents children with an object that should be sorted to match one of two preexisting objects based on either color or shape. A total score is again calculated based on accuracy and speed. The List Sorting Working Memory Test captures working memory by presenting pictures of food or animals accompanied by their names to participants. Afterwards, participants are asked to repeat the items in order from smallest to largest. Task difficulty is modulated by progressively increasing the number of items and categories (animals and food). For more details on the tasks, please refer to [44]. All indicators of SR that were considered in the measurement of SR and outcome variables are listed in Table 1.

Individual scale scores from these instruments were categorized based on the dual-process theory as outlined in [4]. Measures capturing rapid, bottom-up responses to high incentive stimuli – such as sensation seeking, fun seeking, reward responsiveness and drive – alongside emotion-laden impulsivity (positive and negative urgency, inhibition) were classified as components of the hot SR system. In contrast, indicators of deliberate, top-down processes including planning, perseverance, and attention problems were attributed to the cool SR system. Given their reliance on cognitive control mechanisms, the three EF indicators were categorized as part of the cool component.

Some indicators of SR were not included in our measurement model: Firstly, the assessment protocol of the ABCD study included three tasks (Monetary Incentive Delay task, Stop-Signal task, and Emotional N-Back task) related to SR that were administered in a magnetic resonance imaging (MRI) scanner, while functional MRI data was collected. They were excluded from the present analysis due to the situational differences between these tasks and the rest of the indicators of SR, as well as limited value of some of the behavioral data (see, e.g., [45] for details). Secondly, a cash choice task was included in the ABCD protocol as a delay of gratification indicator. Although delayed discounting tasks are considered to measure SR, we did not include this task in the measurement model of SR because of its low discrimination power and lack of correlation with any other indicator of SR. Furthermore, there were conceptual flaws with the task. In contrast to previous studies showing an association between performance on a cash choice task and externalizing behavior [46], participants in the ABCD study could not expect a real payment. The cash choice task was thus only imaginary and as such potentially less sensitive to measuring individual differences in delay discounting. Grades were retrieved from the Kiddie-Structured Assessment for Affective Disorders and Schizophrenia (KSADS-5, [47,48]).

## Statistical analysis

SR is commonly conceptualized as a multifaceted construct comprising domain-specific hot and cool processes. Guided by this dual-process theory, we conducted confirmatory factor analyses (CFAs) to estimate the three previously postulated models. CFAs were performed with the lavaan package [50] in the R Software for Statistical Computing (version 4.2.0, [51]). Please note that despite the confirmatory nature of the analyses, we also employed an exploratory approach to compare the three defensible models and determine the best fit for the data within the dual-process framework. Firstly, univariate distributions of the variables were visually inspected, but no univariate outliers were detected. Therefore, we assumed that the multivariate distributions were neither affected by outliers. The univariate distributions of the CBCL attention scale was skewed, thus the maximum likelihood estimator with robust standard errors was applied in the factor analysis. Given 1.3% missing values, the full information maximum likelihood estimator of lavaan was used. Latent variables in all three defensible models were scaled with reference indicators ($G_{SR}$: reward responsivity, Hot: positive urgency, Cool: planning, EF: flanker task) and the variance of the factors was freely estimated.

**Table 1. Indicators and outcomes of self-regulation in the ABCD dataset.**

| Test (ABCD dataset name) | ABCD variable name | Assessment domain | Assessment modality | SR system | Reliabilities | Scores |
|---|---|---|---|---|---|---|
| Indicators of SR | | | | | | |
| Child Behavior checklist (CBCL) (abcd_cbcls01) | cbcl_scr_syn_attention_r | Attention problems | Caretaker-report | Cool | $r$'s = .82-.95* | Raw scores. 3-point Likert scale. Higher scores indicate less behavioral regulation. Scores were reversed for the present analysis to indicate better self-regulation at higher levels. |
| UPPS_P (abcd_mhy02) | upps_y_ss_lack_of_perseverance | Lack of perseverance | Self-report | Cool | $r$'s = .64 -.83** | 4-point Likert scale. Higher scores indicate more impulsive (less self-regulated) behavior. Scores were reversed for the present analysis to indicate better self-regulation at higher levels. |
| | upps_y_ss_lack_of_planning | Lack of premeditation | | Cool | $r$'s = .66 -.81** | |
| | upps_y_ss_negative urgency | Negative urgency | | Hot | $r$'s = .70 -.86** | |
| | upps_y_ss_positive_urgency | Positive urgency | | Hot | $r$ = .85** | |
| | upps_y_ss_sensation_seeking | Sensation seeking | | Hot | $r$'s = .81 -.93** | |
| Behavioral Inhibition/Behavioral Approach System Scale (BIS BAS scale) (abcd_mhy02) | bis_y_ss_bis_sum | Inhibition | Self-report | Hot | $r$'s = .64-.81*** | 4-point Likert scale. Higher scores indicate less favorable performance. Scores were reversed for the present analysis to indicate better self-regulation at higher levels. |
| | bis_y_ss_bas_fs | Fun seeking | | Hot | $r$'s = .49-.69*** | |
| | bis_y_ss_bas_rr | Reward responsiveness | | Hot | $r$'s = .44-.59*** | |
| | bis_y_ss_bas_drive | Drive | | Hot | $r$'s = .46-.66*** | |
| ABCD Youth NIH TB Summary Scores (abcd_tbss01) | nihtbx_flanker_uncorrected | Flanker task: Cognitive control | Behavioral task | Cool | ICC = 0.92**** | Uncorrected composite score integrating speed and accuracy. Higher values indicate better performance. |
| | Dimensional Card sort tasknihtbx_cardsort_uncorrected | Dimensional Card Sort Task: Set shifting, flexible thinking | | Cool | ICC = 0.92**** | |
| | nihtbx_list_uncorrected | List Sort Working Memory Test: Working memory | | Cool | ICC = 0.86**** | Scores indicate the sum of correct responses over all trials. Higher values indicate better performance. |
| Outcome variables | | | | | | |
| ABCD Parent Diagnostic Interview for DSM-5 Background Items Full (KSADS-5) (dibf01) | kbi_p_grades_in_school | Academic achievement (grades) at baseline | Caretaker-report | | | 1 = A's/ Excellent; 2 = B's/ Good; 3 = C's/ Average; 4 = D's/ Below Average; 5 = F's/ Struggling a lot; 6 = ungraded; -1 = Not applicable |

*(Continued)*

**Table 1.** (Continued)

| Test (ABCD dataset name) | ABCD variable name | Assessment domain | Assessment modality | SR system | Reliabilities | Scores |
|---|---|---|---|---|---|---|
| ABCD Parent School Attendance and Grades (abcd_saag01) | sag_grade_type | Academic achievement (grades) over the last year assessed at two years follow-up assessment | Caretaker-report | | | 1 = A+/ Exceeds excellent; 2 = A/Excellent; 3 = A-/ Approaching Excellent; 4 = B+/Very Good; 5 = B/Very Good; 6 = B-/Approaching Very Good; 7 = C+/Good+; 8 = C/Good; 9 = C-/Approaching Good; 10 = D+/ Satisfactory; 11 = D/Sufficient; 12 = F/Standards Fail; -1 = Not Applicable; 777 = Refuse to answer |

Parenthetical dataset names in the first column indicate the ABCD dataset-derived short names of the data structure from which the data was retrieved. Reliabilities:

*1-week test-retest reliability [49],

**3–4-week's test-retest [31],

***8-weeks' to 1-year test-retest reliability [31],

****[44]

We additionally assessed the predictive validity of the factors on grades at the baseline assessment, as well as two years later. Since the outcome variables are ordinal, prediction models were estimated with the weighted least square mean and variance adjusted estimator of lavaan. Missing values were treated with pairwise deletion in this case. We further split the data into five subsets and performed a cross validation (CV) to test the robustness of the factor structure in smaller samples and across sampling variations. To account for the nested structure of the data due to family membership, the full sample was first split between participants with and without family members in the study. Participants were then randomly allocated to the CV sets ensuring each set contained only one member per family. This approach aimed to mitigate potential bias in the estimated parameters' standard errors arising from family membership-related dependencies within the data. The procedure was repeated with 50 CV sets.

Model fit was assessed with alternative indices of fit, given the high sensitivity of the $\chi^2$-test to large samples as the present one. We interpreted model fit as satisfactory if the root mean square error of approximation (RMSEA) <.07 and the standardized root means square residual (SRMR) <.08. Additionally, the comparative fit indices (CFI) should be > .90 [52]. Given the very large sample size, results were interpreted in terms of effect sizes and not only statistical significance. The guidelines by Cohen (1988) were applied ($r$ > .1 as small, $r$ > .3 as medium, and $r$ > .5 as large effect; [53]).

## Data and code availability

This study was not preregistered. The data used in this research were sourced from the ABCD study (https://abcdstudy.org), hosted within the NIMH Data Archive. Due to data access restrictions outlined by the ABCD study, the authors are unable to publicly share the study's dataset. However, the ABCD study embraces an open science approach and data access can be requested by any eligible researcher with a valid research use of the data (https://nda.nih.gov/nda/access-data-info). Please note that the full analysis code can be accessed at https://osf.io/nzjf7/.

## Results

### Defensible models of self-regulation as measured in the ABCD dataset

We tested the three proposed defensible models described above. Model 1 included a single general SR factor ($G_{SR}$) capturing the variance shared by all indicators, as well as three orthogonal nested factors Hot, Cool, and EF. Model fit was not satisfactory (CFI = .890, RMSEA = .068, SRMR = .05). $G_{SR}$ was mostly determined by emotional (positive and negative) urgency ($\lambda_{GSR}$ = .54-.70), but also planning and perseverance ($\lambda$ = .29-.31). Loadings of attention, EF, and sensation

seeking on $G_{SR}$ were significant, but weak ($\lambda$=.11-.19). The reference indicator (reward responsivity) loaded substantially, but negatively onto the factor ($\lambda$=-.18). All loadings onto the Hot factor were moderate to high with more variance shared across BIS/BAS scales ($\lambda$=.38 -.73) as compared to UPPS-P scales ($\lambda$=.29 -.36). The Cool factor accounted for shared variance across all its indicators, i.e., perseverance ($\lambda$=.72), planning ($\lambda$=.50), and attention ($\lambda$=.23). The EF factor also explained considerable variance in its three indicators ($\lambda$=.46-.69). The variances of Hot, Cool, and EF were substantial ($SD_{Hot}$=1.06, $SD_{Cool}$=1.20, $SD_{EF}$=5.75, $p$<.001), but the variance of $G_{SR}$ was smaller ($SD_{GSR}$=.52, $p$=.04). Since $G_{SR}$ mainly explained variance in UPPS-P scales, we added a method factor to capture survey-specific variance of the BIS/BAS scales. After doing so the Hot factor could not be established anymore ($SD_{Hot}$=.46, $p$=.485) and $G_{SR}$ mainly explained variance in hot indicators ($\lambda_{hot\_indicators}$=.19 -.68; $\lambda_{cool\_indicators}$=.12 -.23; $\lambda_{EF\_indicators}$=.07 -.14). We, therefore, concluded that despite the large sample size, $G_{SR}$ is rather unstable. In conclusion, in Model 1, the loading patterns of the three nested factors were plausible and substantial, but the strength of the domain-general factor $G_{SR}$ was insufficient for this general factor to be established.

Since the assessment modality was shown to play a crucial role in data-driven modeling approaches of SR [24], we next changed Model 1 by extracting EF from the model. Thus, in Model 2, additionally to $G_{SR}$, we estimated Hot and Cool orthogonal nested factors (Model 2a), but EF was estimated separately (Model 2b). The fit for Model 2a was slightly improved as compared to Model 1 (CFI=.903, RMSEA=.082, SRMR=.047). Note that these models can only be descriptively compared as they are not nested in each other. The general loading pattern of $G_{SR}$ in Model 2a was similar to Model 1 with the two emotional urgency scales loading most strongly ($\lambda$=.48-.62), followed by planning and perseverance ($\lambda$=.32-.33,). The only BIS/BAS scale that loaded onto $G_{SR}$ was the reference indicator reward responsivity, albeit negatively ($\lambda$=-.3). The loading of the attention scale was again substantial but weak ($\lambda$=.15). The general pattern for the Hot and Cool factors was similar to Model 1. Even though the variance of $G_{SR}$ was significant ($SD$=.87, $p$<.001), the same loading pattern and a $G_{SR}$-Hot trade-off was observed, similarly to Model 1. Model 2b was just identified with three indicators. Thus, the fit cannot be assessed. To obtain an estimate of model fit, a fourth indicator – the attention scale from the CBCL – was added to the model. The fit was very good (CFI=.994, RMSEA=.035, SRMR=.011). Factor loadings on EF were similar to Model 1 ($\lambda$=.47-.71). Adding the attention scale score ($\lambda$=.22) did not alter these loadings significantly.

In Model 3a, the general factor $G_{SR}$ was omitted from the model and indicators of EF were included. In contrast to previous models, correlations between Hot, Cool and EF were freely estimated. The fit of Model 3a was not satisfactory (CFI=.744, RMSEA=.095, SRMR=.073). All factor loadings on the Hot factor were significant with medium to large magnitude. The strongest loadings were of fun seeking ($\lambda$=.72), drive ($\lambda$=.63), and reward responsivity ($\lambda$=.62) from the BAS scale. The UPPS-P emotional urgency scales ($\lambda$=.40-.44), inhibition ($\lambda$=.38), and sensation seeking ($\lambda$=.32) also had substantial factor loadings. The Cool factor showed medium to strong factor loadings ($\lambda$=.29-.80), as did the EF factor ($\lambda$=.48-.72). EF was correlated with the Hot factor ($r$=.15) and the Cool factor ($r$=.12), but Hot and Cool showed no association ($r$=-.02). All factor variances were significant ($p$<.001). In sum, this model showed meaningful factor loadings, but unsatisfactory model fit. Given the well-known biases associated with responding on self-report questionnaires (e.g., [54]), this finding is not surprising. In the next step we, therefore, adjusted the model to improve its fit.

Firstly, we extended Model 3a by adding an orthogonal method factor (M) capturing variance attributed to the type of survey, namely the BIS/BAS questionnaire (Model 3b; CFI=.86, RMSEA=.072, SRMR=.059). BIS/BAS scales loaded on M with medium to high magnitude ($\lambda$=.31-.75). Consequently, factor loadings of BIS/BAS scales on the Hot factor were reduced ($\lambda$=.19-.39) and the Hot factor mainly explained variance in the two emotional urgency UPPS-P scales ($\lambda$=.64-.78), but also sensation seeking ($\lambda$=.26). The general pattern of Cool factor loadings persisted ($\lambda$=.32-.67) and EF factor loadings remained unchanged. Interestingly, a correlation between the Hot and Cool factor emerged ($r$=.33) in addition to the correlation between Hot and EF ($r$=.19), but the association between Cool and EF slightly decreased ($r$=.11). Next, we consulted modification indices and added residual correlations to the model (Model 3c). Adding residual correlations between inhibition and positive ($r$=.61) and negative ($r$=.55) urgency; sensation seeking with fun seeking ($r$=.24);

planning with reward responsivity ($r=-.25$) and attention ($r=-.38$); and lastly perseverance with fun seeking ($r=-.16$), sensation seeking ($r=-.17$) and reward responsivity ($r=-.20$) improved the model fit sufficiently (CFI = .945, RMSEA = .049, SRMR = .041). Residual correlations indicate the amount of shard variance between two indicators beyond their commonality accounted by the factors, i.e., beyond the variance that is shared among all indicators. The strongest residual correlations between inhibition and emotional urgency can be explained by the fact that items on these scales are phrased negatively (e.g., for positive urgency: '*When I am in a great mood, I tend to do things that can cause me problems*' and for inhibition: '*I worry about making mistakes.*'). Other scales are coded in the same direction, but phrased in a positive way. The association might, therefore, stem from a negative self-image. Small negative correlations between cool (lack of planning and perseverance) and hot indicators (reward responsivity, fun seeking, sensation seeking) beyond the factors could be explained by self-report biases towards extreme answers. Whether these residual correlations are robust across smaller samples was tested in the cross validation. Notably, after adding residual correlations, the inhibition scale loaded on the Hot factor with weakly negative strength ($\lambda=-.16$). Given the model fit and loading patterns, Model 3c was retained for further analysis. The correlation between Hot and Cool was of medium strength ($r=.34$), while Hot and EF correlated weakly ($r=.17$) and EF and Cool did not correlate ($r=.07$). Fit indices for all defensible models are summarized in Table 2 and factor loading are visualized in the Supplementary Material (S1 Fig). The final model is depicted in Fig 2A.

### Predictive power of the factors for academic achievement

We then tested the predictive validity of SR, as measured in Model 3c (i.e., excluding $G_{SR}$, but including EF), for a daily-life outcome. Academic success as indicated by grades at the time of the SR assessment (baseline) and grades two years later were included as endogenous variables in Model 4. The model fit was satisfactory (CFI = .939, RMSEA = .061, SRMR = .046). Grades at baseline and two years later were correlated ($r=.46$). SR factors jointly explained 26% of the variance in the grades at baseline, and 27% of the variance two years later. EF and Cool contributed to the overall prediction at both times ($\beta_{EF}=.33-.35$, $\beta_{Cool}=.32$), which highlights the relevance of EF measures as well as survey-based cognitive control for school performance. The Hot factor did not contribute to the prediction of school performance ($\beta_{Hot}=.01-.05$). In sum, the predictive power of the factors for academic success was highlighted by the data. Model 4 is illustrated in Fig 2.

### Robustness of the model

To test the robustness of the factor structure of Model 3c in smaller, family-independent samples and across sampling variations we conducted a five-fold CV ($N_{CV1-5}=2,376$). The results are illustrated in Fig 3. Results of a further trial with a

**Table 2. Model fit of three defensible confirmatory factor analysis (CFA) models.**

| Model | Model description (simplified) | CFI | RMSEA | SRMR |
|---|---|---|---|---|
| 1 | $G_{SR}$ + Hot + Cool + EF | .890 | **.068** | **.050** |
| 2a | $G_{SR}$ + Hot + Cool | **.903** | .082 | **.047** |
| 2b | EF (incl. CBCL attention problems) | **.994** | **.035** | **.011** |
| 3a | Hot + Cool + EF | .744 | .095 | **.073** |
| 3b | Hot + Cool + EF + M | .860 | .072 | **.059** |
| 3c | Hot + Cool + EF + M + residual correlations | **.945** | **.049** | **.041** |
| 4 | Grades at baseline + Grades two years later ~ Hot + Cool + EF + M + residual correlations | **.939** | **.061** | **.046** |

Models 1 to 3c test the measurement model of self-regulation (SR); Model 4 tests the predictive validity of SR factors. Parameters that meet predefined thresholds are printed in bold. $G_{SR}$: general SR factor; Hot/Cool: domain-specific factors for hot/cool components of SR; EF: domain-specific factor of executive functions; CBCL attention problems: attention scale of the Child Behavior Checklist; M: method factor for BIS/BAS scales; CFI: comparative fit index; RMSEA: root mean square error of approximation; SRMR: standardized root mean square residual.

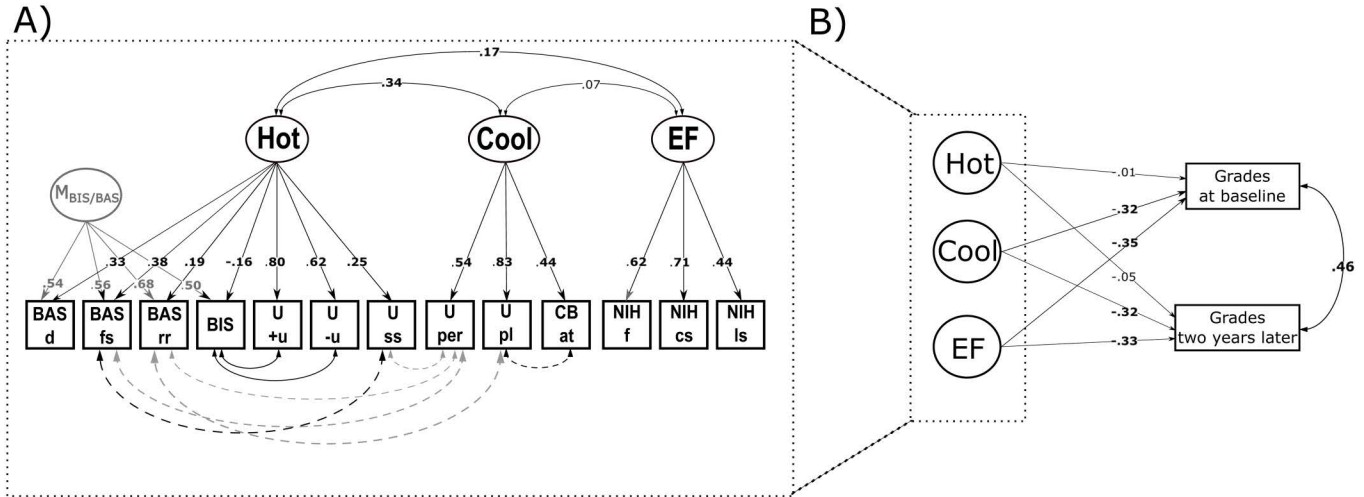

**Fig 2. Best fitting measurement and structural model of self-regulation in the ABCD dataset.** (A) Schematic representation of a measurement model of self-regulation (SR) in the ABCD dataset (Model 3c) with standardized factor loadings. Hot/Cool/EF: domain-specific factors for hot/cool/executive functions components of SR; M_BIS/BAS: method factor for BIS/BAS indicators; U: UPPS-P; +u: positive urgency; -u: negative urgency; ss: sensation seeking; pl: planning; per: perseverance; BIS/BAS: Behavioral Inhibition and Behavioral Approach System Scale, inhibition; rr: reward responsivity; fs: fun seeking; d: drive; CB: Parent Child Behavior Check List; at: attention problems; NIH: NIH toolbox behavioral tasks; f: flanker task; cs: dimensional card sort task; ls: list sort working memory test. Residual correlations: grey color: negative correlation, black color: positive correlation; dashed line: correlation > .3, but < .5; solid line: correlation > .5 (B) Model to test predictive validity of SR on academic success (grades at baseline and grades two years later). Regression weights are semi-standardized on the latent variables' variance.

50-fold CV ($N_{CV1-50}$ = 238) are reported in the Supplementary Material (S2 Fig). The measurement model of SR was robust across CV sets and trials. Therefore, the second hypothesis concerning the robustness of the model in smaller samples was supported here.

## Discussion

In the present study, we confirmed that the extensive ABCD dataset effectively captures aspects of SR consistent with the dual-process theory. We proposed a list of defensible measurement models that can be applied depending on the goal/power of the study. For studying SR in the ABCD dataset, we recommend applying a measurement model including correlated first-order factors with hot, cool and EF components. In this study, the Hot factor describes the ability to maintain control in the presence of strong emotions and high levels of responsivity or drive towards rewards. The Cool and EF factors describe cognitive control, assessed through survey- and behavioral measures, respectively. The EF factor, therefore, does not necessarily represent a trait, but rather a method-specific representation of cool processes. Notably, distinguishing EF and Cool facets as separate factors is appropriate in the ABCD dataset due to the differences in measurement approaches. However, this distinction may not be theoretically generalizable to any psychometric study on SR, where method variance may be better controlled by the use of a more comprehensive multivariate assessment battery.

There has been an ongoing debate about whether these domains should be combined and whether they even represent the same constructs [24,30]. We argue that recognizing the components of SR captured by both behavioral tasks and surveys outweighs the costs of heterogeneous loading patterns among the indicators in this study. Furthermore, a diverse set of indicators is more likely to eliminate construct-irrelevant variation [55]. Given these advantages, alongside our findings (especially the correlations between the factors reported here), we strongly advocate for incorporating Hot, Cool, and EF factors of SR as correlated factors within the same model. We further tested the factors' predictive validity on a relevant daily-life outcome, i.e., academic achievement. Notably, the amount of variance explained in academic

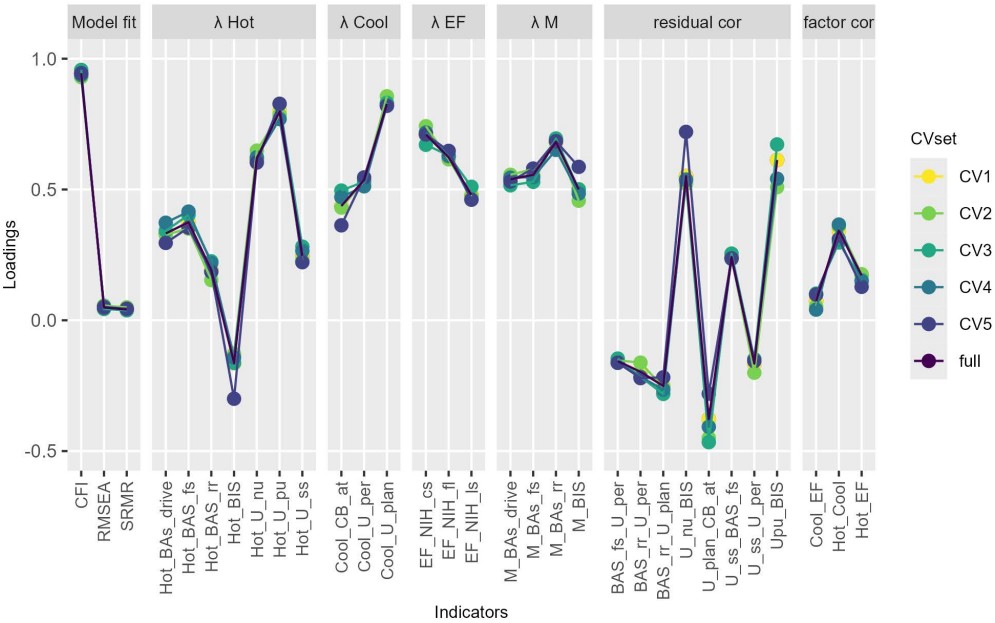

**Fig 3. Five-fold cross validation (CV) of the measurement model of self-regulation.** Factor loadings of the full dataset per factor (full) are compared to factor loadings of cross-validation sets (CV1-CV5) to test the robustness of the factor structure over smaller samples. λ Hot/Cool/EF: factor loadings onto domain-specific factors for hot/cool/executive function components of self-regulation (SR); λ M: factor loadings onto method factor for BIS/BAS scales; residual cor: residual correlations; factor cor: correlations between factors; U: UPPS-P; pu: positive urgency; nu: negative urgency; ss: sensation seeking; pl: planning; per: perseverance; BIS/BAS: Behavioral Inhibition and Behavioral Approach System Scale; rr: reward responsivity; fs: fun seeking; d: drive; CB: Parent Child Behavior Check List; at: attention problems; NIH: NIH toolbox behavioral tasks; f: flanker task; cs: dimensional card sort task; ls: list sort working memory test.

achievement (26–27%) is comparable to or even slightly higher than the amount of variance in academic achievement explained by general intelligence (29%) or intelligence, personality, and emotional intelligence combined (22.9–24.6%) in two meta-analyses [56,57]. We conclude that the model has high predictive validity on academic achievement. Academic achievement is considered to be associated with the Cool and EF factors more strongly as compared to the Hot factor (e.g., [35]). Our study findings support the larger predictive power of these factors in this context. Despite a previously postulated indirect link between the regulation of negative emotions and academic success [58], the Hot factor encompasses the regulation of both positive and negative emotions, as well as various facets of reward responsivity. The scope of the Hot factor might, therefore, be too broad to predict academic achievement. Future studies should additionally take into account outcomes that can be considered as consequences of the hot system of SR. Building upon previous evidence [9], we suggest for future studies, to consider the predictive power of the established SR factors for substance-use. This will most likely be possible as soon as early adulthood age ranges will become available in the longitudinal ABCD study. We assume that reward-related outcomes, such as substance-use behavior would be rather predicted by the Hot factor of SR. Since the model is also robust in smaller samples and independent of family-membership, we conclude that it is well suited to investigate SR in the ABCD dataset.

The current study provides evidence for a measurement model with domain-specific Hot, Cool and EF factors. We additionally addressed the question whether they are connected by a higher-order general factor or whether they are better represented as correlated first-order factors. We show that an integration of a domain-general factor with indicators of task-and survey-based SR in a dual-process model of SR is feasible, however it should be applied with caution. In the present analysis, the inclusion of more diverse indicators weakened the strength of $G_{SR}$ and blurred its distinction from

the Hot factor. Depending on the research question and study design – for instance when only either typical survey-based or maximal task-based SR measured are relevant, and domain-general processes are the focus - a hierarchical model may be appropriate. However, we would not recommend its use universally. Considering the weak factor loadings of most indicators onto $G_{SR}$ in our study in combination with a recent study that did not support the idea of a common inhibitory process in emotion regulation and cognitive control [34], we conclude that a domain-general factor capturing hot and cool processes alike, might only be observable in very high-power study designs with extremely large samples, such as the one presented here. Given its instability even in these cases, its interpretability remains difficult. Hence, our findings confirm the challenges of capturing constructs using different assessment modalities and raise questions whether SR is best represented as a general factor - particularly in small sample studies - also beyond the ABCD dataset.

For further use in the ABCD study, we recommend Model 3c that covers diverse indicators but does not estimate a domain-general factor. Instead, correlations between the domain-specific factors can be estimated and interpreted. The Hot factor was correlated with the Cool and EF factors, supporting the approach of estimating both assessment modalities (i.e., survey- and task-based indicators) within the same model. The Cool and EF factors were not substantially correlated. The correlation between Hot and Cool/EF indicates that the ability to inhibit oneself in the light of strong emotions and being less sensitive to fun/reward is related to cognitive control. The finding of an association between Hot with Cool/EF is in line with existing literature on a negative association between impulsivity and EFs [59] supporting the notion that cognitive control plays a key role in rational behavior and decision making in light of strong emotions or anticipated reward. Conversely, cognitive control, as represented by the EF and Cool factors of SR, might also be influenced by the construct captured with the Hot factor. Cognitive control itself can be described as an emotional process, since its application depends on the presence of a conflict that necessarily elicits short, sometimes covert negative emotions [60]. The correlation might also partly be explained by some theoretical overlap between the Hot and Cool factors in terms of the ability to engage control. The hot indicators describe the ability to control oneself when faced with strong affect or in other specified situations (e.g., '*When I am doing well at something, I like to keep doing this*', '*When I am upset I often act without thinking.*', or '*When I am in a great mood, I tend to do things that can cause me problems.*'). The Cool factor also describes control mechanisms, such as inhibition, attention, and planning, however without a specified context (for Cool: e.g., '*I am very careful*' or '*I finish what I start*', '*can't concentrate*'). This dataset-specific representation of the Cool factor in terms of broad, decontextualized and long-term control mechanisms, along with the more obvious issue of method-specificity, may also explain the lack of association between Cool SR and EF, despite their close theoretical link. SR captured by behavioral task performance, as measured with EF, is less closely related to delayed (as captured by the Cool factor) than to immediate SR, as they operate on different temporal scales of SR.

## Limitations

We acknowledge that the design of the assessment protocol in the ABCD study was not primarily designed to psychometrically represent SR within the dual-process theory. The boundaries between subcomponents of SR are often ambiguous due to the lack of a universally accepted definition of its predictors, outcomes and integral concepts. In a study allowing for greater flexibility in variable inclusion, we might have chosen more nuanced indicators of SR in surveys and behavioral tasks to ensure balanced coverage of both hot and cool components. However, the goal of the present study was not to test the dual-process theory, but rather to see whether it is applicable to the ABCD dataset with the resources available. Furthermore, applying the dual-process theory is not the only way of modeling SR (see [4] for a review) and we do not claim superiority of the dual-process model over given alternatives. However, our study supports the relevance of the dual-process model in the context of decision making and brain-behavior relationships, which are specifically addressed in the ABCD dataset.

## Concluding remarks

A common issue in the study of SR is that distinct concepts (e.g., motivation, impulsivity) are often wrongly termed SR, even in cases where domain-generality is by no means empirically covered [5]. To prevent inadequate conclusions made

due to an imbalance in terminology use and construct coverage, we encourage a clear distinction between the study of SR vs. its components in future studies. In the present study, the dual-process model was chosen to theoretically guide the analyses with the goal to establish a model within the ABCD dataset, and even with the potential to be robust for different study designs and large-scale datasets with a comparable set of variables. For future applications, we provide analysis code and suggest classifying indicators as part of either the hot (including an emotional/reactive context) or cool (including cognitive control/reflective processes) components of SR. Since this is easily implemented, and neural correlates for these components have been established, with the current study we hope to foster consistent future research about SR and its neural correlates in the ABCD dataset and beyond.

## Supporting information

**S1 Fig. Measurement models for all defensible self-regulation models.** Schematic representation of all three defensible measurement models of SR with standardized factor loadings. Reference indicators are depicted with dashed lines. (A) Model 1 (B) Models 2a and 2b (C) Model 3b; Hot/Cool/EF: domain-specific factors for hot/cool/executive functions components of SR; $M_{BIS/BAS}$: method factor for BIS/BAS indicators; U: UPPS-P; +u: positive urgency; -u: negative urgency; ss: sensation seeking; pl: planning; per: perseverance; BIS/BAS: Behavioral Inhibition and Behavioral Approach System Scale, inhibition; rr: reward responsivity; fs: fun seeking; d: drive; CB: Parent Child Behavior Check List; at: attention problems; NIH: NIH toolbox behavioral tasks; f: flanker task; cs: dimensional card sort task; ls: list sort working memory test.
(TIF)

**S2 Fig. 50-fold cross validation of the best fitting measurement model of self-regulation.** Factor loadings of the full dataset per factor (full) are compared to factor loadings of cross-validation sets (CV 1-CV 50) to assess the robustness of the factor structure over smaller samples. Fit: model fit; load_Hot/Cool/EF: factor loadings onto domain specific factors for hot/cool/executive function components of self-regulation (SR); load_M: factor loadings onto method factor for BIS/BAS scales; res_cor: residual correlations; factor_cor: correlations between factors; U: UPPS-P; +u: positive urgency; -u: negative urgency; ss: sensation seeking; pl: planning; per: perseverance; BIS/BAS: Behavioral Inhibition and Behavioral Approach System Scale; rr: reward responsivity; fs: fun seeking; d: drive; CB: Parent Child Behavior Check List; rb: rule breaking behavior; ag: aggressive behavior; th: thought problems; at: attention problems; NIH: NIH toolbox behavioral tasks; f: flanker task; cs: dimensional card sort task; ls: list sort working memory test.
(TIF)

## Author contributions

**Conceptualization:** Merle Johanna Marek, Andrea Hildebrandt.

**Data curation:** Merle Johanna Marek.

**Formal analysis:** Merle Johanna Marek.

**Methodology:** Merle Johanna Marek, Andrea Hildebrandt.

**Software:** Merle Johanna Marek, Andrea Hildebrandt.

**Supervision:** Axel Heep, Andrea Hildebrandt.

**Visualization:** Merle Johanna Marek.

**Writing – original draft:** Merle Johanna Marek.

**Writing – review & editing:** Axel Heep, Andrea Hildebrandt.

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
