## [Decision Letter · Decision Letter 0]

10 Dec 2024

PONE-D-24-12171The measurement of self-regulation in the Adolescent Brain Cognitive Development (ABCD) StudyPLOS ONE

Dear Dr. Marek,

Thank you for submitting your manuscript to PLOS ONE. After careful consideration, we feel that it has merit but does not fully meet PLOS ONE’s publication criteria as it currently stands. Therefore, we invite you to submit a revised version of the manuscript that addresses the points raised during the review process.

We look forward to receiving your revised manuscript.

Kind regards,

Iftikhar Ahmed Khan

Academic Editor

PLOS ONE

**Journal Requirements:**

4. We notice that your supplementary figures are uploaded with the file type 'Figure'. Please amend the file type to 'Supporting Information'. Please ensure that each Supporting Information file has a legend listed in the manuscript after the references list.

Reviewers' comments:

Reviewer's Responses to Questions

**Comments to the Author**

1. Is the manuscript technically sound, and do the data support the conclusions?

Reviewer #1: Yes

2. Has the statistical analysis been performed appropriately and rigorously?

Reviewer #1: Yes

3. Have the authors made all data underlying the findings in their manuscript fully available?

Reviewer #1: No

4. Is the manuscript presented in an intelligible fashion and written in standard English?

Reviewer #1: Yes

5. Review Comments to the Author

**Reviewer #1:**  The submitted manuscript reports an interdisciplinary approach to research related to self-regulation dealing with significant assessment issues. The aim of developing a measurement model for assessing self-regulation within the ABCD study could have large impact because of the widespread use of ABCD data in brain research. Because I am no expert in the field of neuroscience, my recommendations mainly refer to a differential/assessment perspective and to issues of research on self-regulation in general.

However, some aspects of the manuscript could be improved as explained in the following:

- Structure: from a theoretical perspective, the paper is about the valid assessment of self-regulation in a specific data repository considering instruments implemented there. Because the basis of assessment should be the theoretical conceptualization of a construct, I suggest to focus on the conceptualization of SR first, before introducing the ABCD study more in detail as context for the current research project. In some parts of the introduction, the structure could be improved with subheadings referring to self-regulation and the ABCD study as well as with self-regulation tasks within the ABCD study ("Present study"). Currently, those contents are partly mixed up what can lead to reduced comprehension by readers.

- in the introduction, the motivation for/implications of studying brain-behavior associations should become clear because this is one important context and starting point of the reported research.

- l.43: a priori

- l. 47/48: the sentence confuses two aspects: validity of used instruments and measurement models for representing latent constructs. Both are important issues when brain-behavior associations (and psychological variables in general) are examined

- l. 48: I suggest to use "research on" instead of "tests"

- l. 50 - l. 54: These sentences use present tense. For consistency when referring to the reported research, past tense should be used. Additionally, to me the intention of this part was not completely clear. If the authors wanted to briefly outline the aims of the reported research, I recommend to clearly state that they are writing about the aims. Taking into account my suggestion referring to the structure of the introduction this would be placed after referring to self-regulation in the introduction.

- l. 55: for a sound introduction, for me, an overview about definitions and the nomological network of self-regulation was missing when it was introduced. what are commonalities of definitions and what are shortcomings the authors write about? This could include aspects following later, e.g., in ll. 84

- l. 57: "biological-" and so on should be written without hyphen

- l. 61: for readers it would be helpful to define what "context" means and to refer also to the role individual goals have related to self-regulation

- l. 62: for understanding it would be beneficial to explain the relation between self-control (probably referred to by the authors when writing about behavioral conflict) and self-regulation

- l. 65: when referring to the dual-process theory the authors name domain-specific facets which could be understood in a misleading way. maybe it would be clearer to refer to classes of processes or to two systems complementing each other

- l. 67: because EF are central to this research, the three EF introduced should be explained briefly

- when referring to measurement approaches and the structure of self-regulation, previous research on convergent and discriminant validity should be considered, because differences between tasks refer not only to kind of measure but also to the subconstruct measured, see e.g., Duckworth & Kern (2011) https://doi.org/10.1016/j.jrp.2011.02.004

- l. 91: "content validity" - I guess, the authors mean construct validity?

- l. 97: the authors should briefly explain why behavioral (experimental?) tasks are advantageous despite low reliability, which theoretically implies low validity, too. They should explain why they are better suited for studying brain-behavior relationships.

Present Study: The authors could argue more transparently why they decided to test the three models and what tasks were considered to load on them. From a theoretical point of view, I have not completely understood, why EF were considered to mirror a factor distinct from (hot)/cool system on the same hierarchical level. Therefore, naming example tasks would be helpful when introducing EF and hot/cool processes above.

 l. 137 referring to different assessment domains is not clear to readers who do not know the ABCD tasks because EF and cool processes are often measured as behavioral tasks.

For introducing tested models 1 to 3 a figure displaying them would improve the clarity of this part.

Methods:

- l. 180: what is the reference [4] referring to? On what basis did the authors choose tasks potentially measuring self-regulation. For transparency and reproducibility of results more details are needed.

- Similarly, more explanation is needed, on what basis have the authors decided what instruments are used as indicators for hot/cool system/EF?

- l. 184: UPPS-P is an abbreviation that has not been introduced to readers.

- in general, the usage of many abbreviations for considered tasks are not easy to follow. If future readers should mainly consist of researchers familiar with ABCD and those abbreviations are common in this context, they could remain.

- Tasks considered in this research should be described briefly in text or in table 1 with example tasks/items, number of tasks/items etc.

- Do abbreviations in Table 1 refer to variable names used in ABCD data? otherwise, they are rather confusing. Abbreviations such as "CB_at" should be left out in table 1 or explained in a table note because acc. to APA table content should be understood on its own so that abbreviations have to be explained again in each table.

- ll. 231: What were criteria for splitting the sample into subsamples, were them randomly assigned to subsamples besides considering family membership?

- in general, introducing subheadings for the description of sample, procedure, instruments, and statistical analysis would improve the structure of the method section.

Results:

- l. 254: "is mostly determined" should be past tense

- Capturing the results at first glance would be supported by providing figures for all model results including path coefficients (maybe as appendix)

Discussion:

- Because the authors repeatedly refer to differences in assessment (self-report vs. behavorial tasks) influencing correlational findings I recommend to discuss to what extent do the models and respective factors represent latent constructs in a narrower sense and method factors potentially qualifying the distinction of three dimensions and EF separately from Hot/Cool processes on a construct level.

 although one central aim of the research was to identify a feasible measurement model in the ABCD context, the discussion should refer both to the implementation in research using ABCD data and to implications for research on EF/self-regulation in general together with respective assessment issues.

To some extent the finding of no general SR-factor represents the heterogeneity of approaches on self-regulation.

general remarks:

- the authors write about cold vs. hot processes/functions. For example, in Metcalfe & Mischel (1999) the common term is rather "cool" EF than "cold" EF. Following this, I recommend to change "cold" into "cool".

- sometimes, statements referring to previous research results have no reference, e.g. in

- some English phrases seem a little bit uncommon, e.g.,: l. 72, l. 85, l.96-99

"exploit these data sets" in l. 43. Hence, I recommend to check English language again during revision.

- I recommend to check what abbreviations are necessary and support text fluency and what abbreviations could be omitted to ensure that readers can easily understand the meaning of text content.

6. PLOS authors have the option to publish the peer review history of their article (what does this mean? ). If published, this will include your full peer review and any attached files.

**Do you want your identity to be public for this peer review?** For information about this choice, including consent withdrawal, please see our Privacy Policy .

Reviewer #1: **Yes: ** Julia Grass

---

## [Author Response · Author response to Decision Letter 1]

17 Jan 2025

Dear Prof. Dr. Iftikhar Ahmed Khan, dear Reviewer Dr. Grass,

Thank you for handling and reviewing our manuscript, number PONE-D-24-12171, and for your insightful and positive feedback! We appreciate the constructive comments on how to improve the manuscript. We have carefully considered all comments and incorporated them into the revised manuscript. A version of the revised manuscript without track changes is also provided. Please find below a point-by-point response to the reviewer’s comments. Line numbers refer to the final manuscript without track changes.

Yours sincerely,

Merle Marek, Axel Heep and Andrea Hildebrandt

Journal Requirements:

Please ensure that your manuscript meets PLOS ONE's style requirements, including those for file naming. The PLOS ONE style templates can be found at https://journals.plos.org/plosone/s/file?id=wjVg/PLOSOne_formatting_sample_main_body.pdf and https://journals.plos.org/plosone/s/file?id=ba62/PLOSOne_formatting_sample_title_authors_affiliations.pdf2. We note that the grant information you provided in the ‘Funding Information’ and ‘Financial Disclosure’ sections do not match.

We sincerely thank the editor for the valuable and constructive comments. In response, we have adjusted the size of the figures and updated the panel labels in the figure legends as suggested. Additionally, we have modified the settings with PACE, reformatted the tables, and aligned the title page with the journal’s requirements. We appreciate the opportunity to improve the manuscript and are confident these adjustments enhance its clarity and presentation.

Thank you for pointing this out. We understand the confusion regarding the grant information. To clarify, the authors of this study did not receive any direct grant funding; however, the ABCD study, from which data were utilized, did. Upon resubmission, we will adapt the funding section to include the relevant grant numbers for the ABCD study, ensuring accurate attribution.

We have revised the statement and added a statement in line 330-335 of our manuscript:

‘Due to data access restrictions outlined by the ABCD study, the authors are unable publicly share the study’s dataset. However, the ABCD study embraces an open science approach and data access can be requested by any eligible researcher with a valid research use of the data. Please note that the full analysis code can be accessed at https://osf.io/nzjf7/?view_only=ebd4dfaa5c8b4f0fb499b6071f43831d .’

We appreciate your guidance on this matter and are confident this revised statement adheres to the journal’s policy while providing clear instructions for researchers seeking data access.

4. We notice that your supplementary figures are uploaded with the file type 'Figure'. Please amend the file type to 'Supporting Information'. Please ensure that each Supporting Information file has a legend listed in the manuscript after the references list.

Thank you for bringing this to our attention. We have addressed this by uploading the figures as 'Supporting Information,' ensuring that each supplementary figure is accompanied by a detailed legend within the manuscript.  

Reviewers' comments:

Reviewer's Responses to Questions

Comments to the Author

1. Is the manuscript technically sound, and do the data support the conclusions?

Reviewer #1: Yes

2. Has the statistical analysis been performed appropriately and rigorously?

Reviewer #1: Yes

3. Have the authors made all data underlying the findings in their manuscript fully available?

Reviewer #1: No

Please refer to our previous response to the editor’s third comment regarding data sharing issues.

4. Is the manuscript presented in an intelligible fashion and written in standard English?

Reviewer #1: Yes

5. Review Comments to the Author

Reviewer #1: The submitted manuscript reports an interdisciplinary approach to research related to self-regulation dealing with significant assessment issues. The aim of developing a measurement model for assessing self-regulation within the ABCD study could have large impact because of the widespread use of ABCD data in brain research. Because I am no expert in the field of neuroscience, my recommendations mainly refer to a differential/assessment perspective and to issues of research on self-regulation in general.

However, some aspects of the manuscript could be improved as explained in the following:

- Structure: from a theoretical perspective, the paper is about the valid assessment of self-regulation in a specific data repository considering instruments implemented there. Because the basis of assessment should be the theoretical conceptualization of a construct, I suggest to focus on the conceptualization of SR first, before introducing the ABCD study more in detail as context for the current research project. In some parts of the introduction, the structure could be improved with subheadings referring to self-regulation and the ABCD study as well as with self-regulation tasks within the ABCD study ("Present study"). Currently, those contents are partly mixed up what can lead to reduced comprehension by readers.

We sincerely thank the reviewer for the valuable feedback, particularly regarding the structure and theoretical framing of the introduction. In response to the suggestions, we have revised the structure to improve the flow and added subheadings to enhance clarity. These subheadings now highlight the conceptualization of self-regulation, general measurement issues, and the specific context of self-regulation in the ABCD dataset.

To address the reviewer's concern, we reorganized the paragraph ‘The measurement of self-regulation’ (lines 111–132) so that the ABCD study is discussed only at the end of the paragraph (lines 127–132). This change serves as a bridge to the following section, ‘The measurement of self-regulation in the ABCD study’ clarifying that the initial discussion focuses on measurement challenges in general. Additionally, we now introduce the measures of self-regulation in the ABCD study earlier in the manuscript (lines 147–153, within the paragraph ‘The measurement of self-regulation in the ABCD study’). This is to clarify the selection of the defensible models and to enhance the narrative's coherence. The new text reads:

‘We evaluated all available instruments of the ABCD study related to mental health and neurocognition based on their theoretical overlap with the construct of SR or its subconstructs. Potentially related constructs were identified based on [2] and [30], which include a glossary of major terms and mental processes related to SR and an overview of task and survey-based indicators of SR, respectively. We identified three questionnaires measuring hot and cool facets of self-reported impulsivity and inhibition/reward seeking, as well as caregiver-reported attention problems, and three EF tasks as potential indicators of SR.’

We carefully reflected on the suggestion to prioritize the conceptualization of self-regulation before introducing the dataset. This approach is undoubtedly essential to most psychometric studies, where the theoretical framework typically informs the selection of measures. However, our study takes a different perspective: rather than beginning with a theoretical conceptualization and subsequently choosing corresponding indicators to be applied, we were restricted to start with the predefined measures available in the ABCD dataset and our goal was to explore whether they can meaningfully align with the dual process model of self-regulation. We are concerned that adopting a structure where the theoretical framework precedes the dataset could mislead readers to assume that we first conceptualized self-regulation and then derived measures to be used to test the taxonomic/psychometric model. To avoid this potential misunderstanding, we believe it is critical to emphasize from the outset that our focus is on evaluating the suitability of existing measures within the dataset, rather than operationalizing self-regulation beginning from theory. While the suggested structure offers clear advantages in contexts where theoretically derived measurement models allow flexibility in indicator selection, we believe our approach better avoids potential misinterpretation and more accurately reflects the unique goals of our research. We hope these revisions and our rationale address the reviewer's points while maintaining the study's focus.

- in the introduction, the motivation for/implications of studying brain-behavior associations should become clear because this is one important context and starting point of the reported research.

Thank you for pointing out the importance of highlighting the motivation and implications of studying brain-behavior associations in the introduction. To address this, we have added the sentence in lines 49–51:

‘Understanding the link between brain systems and networks and human behavior is central to cognitive-behavioral neuroscience and offers critical insights into mechanisms underlying cognition, personality, and psychiatric conditions.’

- l.43: a priori

We have corrected the typo in line 53.

- l. 47/48: the sentence confuses two aspects: validity of used instruments and measurement models for representing latent constructs. Both are important issues when brain-behavior associations (and psychological variables in general) are examined

Thank you for this careful observation. We agree that our previous phrasing might have suggested that modeling is directly related to validity of tests. Instead of referring to ‘enhancing psychometrically sound research’, we now refer to ‘improve the statistical analysis of brain-behavior associations’ in line 55-59 and improved the sentence structure to enhance the reading flow:

‘As the expertise of researchers analyzing data from these neuroimaging initiatives is often in neuroimaging but less so in psychometrics, proposals for psychometric models assessed for robustness, together with code for implementation, have the potential to improve the statistical analysis of brain-behavior associations.’

- l. 48: I suggest to use "research on" instead of "tests"

We agree with the suggestion, however, this part was deleted from the text due to the previous suggestion.

- l. 50 - l. 54: These sentences use present tense. For consistency when referring to the reported research, past tense should be used. Additionally, to me the intention of this part was not completely clear. If the authors wanted to briefly outline the aims of the reported research, I recommend to clearly state that they are writing about the aims. Taking into account my suggestion referring to the structure of the introduction this would be placed after referring to self-regulation in the introduction.

We have revised this section to use past tense for consistency when referring to the reported research. Additionally, we have explicitly clarified that this part outlines the goals of our study. The updated text (lines 92–63) now reads:

‘The aim of this study was to foster a well-grounded and consistent construct representation of SR in future brain-behavior association studies in the ABCD dataset – today's largest open longitudinal neuroimaging study. To reach this goal, we investigated alternative measurement models and provided code that can be used in future analyses of SR’s neural correlates.’

Regarding the placement of this part, we respectfully refer to our first comment, where we explain our rationale for the structure of the introduction.

- l. 55: for a sound introduction, for me, an overview about definitions and the nomological network of self-regulation was missing when it was introduced. what are commonalities of definitions and what are shortcomings the authors write about? This could include aspects following later, e.g., in ll. 84

We have expanded the introduction with a more elaborate definition of the construct of self-regulation in lines 66–84. This revised section now addresses the commonalities among definitions (lines 66–79), as well as key shortcomings in the field, such as the jingle-jangle fallacy, which often arises in self-regulation research (lines 79–81). We also discuss the diversity of the postulated self-regulation subconstructs (lines 69–73):

‘SR refers to the ability to actively guide oneself toward a desired emotional, behavioral, or cognitive state (e.g., [1]). It involves adapting internal states (i.e. emotions and cognitions) and behaviors to align with contextual demands [2]. Context, in this regard, encompasses situational factors, social expectations, and personal goals. SR is a complex, multidimensional construct that depends on dynamic interactions between traits (e.g., temperament, impulsivity) and states (e.g., hormonal and physiological conditions) and involves various subskills, such as delay of gratification, executive functions (EFs) and self-control (see [2] for a glossary of related terminology). Although SR and self-control are sometimes used interchangeably, they are more accurately understood as distinct but related concepts. Self-control specifically addresses behavioral regulation in the face of conflicts between competing goals, such as resisting short-term rewards to prioritize long-term outcomes. In contrast, SR encompasses a broader set of processes, including planning, goal-setting, monitoring, in addition to the execution of behavior [1,3,4]. Unlike self-control, SR is not necessarily dependent on the presence of a confli

---

## [Decision Letter · Decision Letter 1]

7 Mar 2025

PONE-D-24-12171R1The measurement of self-regulation in the Adolescent Brain Cognitive Development (ABCD) StudyPLOS ONE

Dear Dr. Marek,

Thank you for submitting your manuscript to PLOS ONE. After careful consideration, we feel that it has merit but does not fully meet PLOS ONE’s publication criteria as it currently stands. Therefore, we invite you to submit a revised version of the manuscript that addresses the points raised during the review process.

We look forward to receiving your revised manuscript.

Kind regards,

Iftikhar Ahmed Khan

Academic Editor

PLOS ONE

Journal Requirements:

Reviewers' comments:

Reviewer's Responses to Questions

**Comments to the Author**

1. If the authors have adequately addressed your comments raised in a previous round of review and you feel that this manuscript is now acceptable for publication, you may indicate that here to bypass the “Comments to the Author” section, enter your conflict of interest statement in the “Confidential to Editor” section, and submit your "Accept" recommendation.

Reviewer #1: All comments have been addressed

2. Is the manuscript technically sound, and do the data support the conclusions?

Reviewer #1: Yes

3. Has the statistical analysis been performed appropriately and rigorously?

Reviewer #1: Yes

4. Have the authors made all data underlying the findings in their manuscript fully available?

Reviewer #1: (No Response)

5. Is the manuscript presented in an intelligible fashion and written in standard English?

Reviewer #1: Yes

6. Review Comments to the Author

Reviewer #1: The authors have thoroughly revised and significantly improved the manuscript so that only minor remarks remain.

Open Data statement:

The new statement on data availability is transparent. Two minor remarks: Adding a link to the homepage, where data sharing restrictions by the ABCD-authors and the contact for requesting data for research purposes would be an improvement of transparency/clarity. The osf link is currently the anonymous one – when preparing the manuscript for publication, you should change it into the official one.

Structure:

The revision of the introduction has improved the clarity and comprehensibility a lot. I understand the decision to introduce the ABCD study first bevor writing about self-regulation in general to prevent misleadingly focusing on SR in general. The authors have improved the clarity of the introduction and added important parts concerning study aims and theoretical information (e.g., definition) about core variables.

Theory (Discussion)

The background for the tested models of SR became clearer including both theoretical and method-based reasoning for not including EF into cool functions. Maybe, this aspect would be valuable to include in the discussion briefly because distinguishing EF and cool functions refers to measures used in ABCD but is not theoretically generalizable to SR in general and other studies.

Ll. 49-51: reference missing

l. 58: I would extent the statement on the aim/benefit of the current research: providing a well-founded measurement model is not only advantageous for statistical analysis but also for drawing valid conclusions based on neuropsychological research on the ABCD study.

ll. 113: The usage of “content domain” is slightly misleading although it corresponds with the term content validity. DoG-measures and EF measures refer to slightly different aspects of self-regulation but also to different measurement approaches.

ll. 119: with the explanation of the reviewers I better understand why they have chosen to write about content validity instead of construct validity. However, because including measurement approaches (partly) referring to different construct facets and testing whether the measurement model represents one general latent construct (what somehow does include the idea of convergent validity), at some point they should refer also to construct validity and not only to content validity.

Methods:

Table 1: I recommend to add one brief statement in the table notes explaining that parenthetical information in the firs column refer sth. referring to the ABCD dataset (which is, what I suppose)

Minor remark: “lavaan” in line 308 seems to be written in another font than the rest of the text.

Discussion:

ll. 545-548: hot and cool system do not mainly refer to long vs. short-term orientation but to the extent to which affect/emotions are involved or rather cognitively focused – mainly referring to long-/short term may be misleading because from a conceptual view basic EF are often considered to be part of the cool system without being long-term oriented per se (but of course helpful to reach long-term goals).

7. PLOS authors have the option to publish the peer review history of their article (what does this mean? ). If published, this will include your full peer review and any attached files.

**Do you want your identity to be public for this peer review?** For information about this choice, including consent withdrawal, please see our Privacy Policy .

Reviewer #1: **Yes: ** Julia Grass

---

## [Author Response · Author response to Decision Letter 2]

14 Mar 2025

Dear Prof. Dr. Iftikhar Ahmed Khan, dear Reviewer Dr. Grass,

Thank you for handling and reviewing our manuscript (PONE-D-24-12171R1) and for your insightful and positive feedback. We sincerely appreciate the constructive comments, which have once again helped us to improve the manuscript. We have carefully considered all suggestions and incorporated them into the revised version. Additionally, we have provided a version of the manuscript without the track changes.

Below, you will find our point-by-point response to the reviewers' comments, with line numbers referring to the final manuscript without track changes.

Yours sincerely,

Merle Marek, Axel Heep and Andrea Hildebrandt

Journal Requirements:

We sincerely appreciate the editor’s careful observation. Upon reviewing the reference list, we confirmed that one cited publication had been retracted in the meantime:

Schmidt, H., Daseking, M., Gawrilow, C., Karbach, J., & Kerner auch Koerner, J. (2022). RETRACTED: Self‐regulation in preschool: Are executive function and effortful control overlapping constructs? Developmental Science, 25(6), e13272.

We have removed this reference from our article and replaced it with a more suitable source to support our assumptions:

Allan, N. P., & Lonigan, C. J. (2014). Exploring dimensionality of effortful control using hot and cool tasks in a sample of preschool children. Journal of Experimental Child Psychology, 122, 33-47.

Reviewers' comments:

Reviewer's Responses to Questions

Comments to the Author

1. If the authors have adequately addressed your comments raised in a previous round of review and you feel that this manuscript is now acceptable for publication, you may indicate that here to bypass the “Comments to the Author” section, enter your conflict of interest statement in the “Confidential to Editor” section, and submit your "Accept" recommendation.

Reviewer #1: All comments have been addressed

2. Is the manuscript technically sound, and do the data support the conclusions?

Reviewer #1: Yes

3. Has the statistical analysis been performed appropriately and rigorously?

Reviewer #1: Yes

4. Have the authors made all data underlying the findings in their manuscript fully available?

Reviewer #1: (No Response)

5. Is the manuscript presented in an intelligible fashion and written in standard English?

Reviewer #1: Yes

6. Review Comments to the Author

Reviewer #1: The authors have thoroughly revised and significantly improved the manuscript so that only minor remarks remain.

We thank the reviewer for the positive evaluation of our revision.

Open Data statement:

The new statement on data availability is transparent. Two minor remarks: Adding a link to the homepage, where data sharing restrictions by the ABCD-authors and the contact for requesting data for research purposes would be an improvement of transparency/clarity. The osf link is currently the anonymous one – when preparing the manuscript for publication, you should change it into the official one.

We have added a link to the webpage providing information on data access in line 339 and have also updated the OSF link.

Structure:

The revision of the introduction has improved the clarity and comprehensibility a lot. I understand the decision to introduce the ABCD study first bevor writing about self-regulation in general to prevent misleadingly focusing on SR in general. The authors have improved the clarity of the introduction and added important parts concerning study aims and theoretical information (e.g., definition) about core variables.

We appreciate your positive feedback on the revised introduction and are pleased that the restructuring and additional theoretical information have improved clarity and are now more in line with the aims of the study. Thank you for your thoughtful review and valuable suggestions, which have helped us to strengthen the manuscript.

Theory (Discussion)

The background for the tested models of SR became clearer including both theoretical and method-based reasoning for not including EF into cool functions. Maybe, this aspect would be valuable to include in the discussion briefly because distinguishing EF and cool functions refers to measures used in ABCD but is not theoretically generalizable to SR in general and other studies.

We have added such a discussion in lines 485-489:

‘Notably, distinguishing EF and Cool facets as separate factors is appropriate in the ABCD dataset due to the differences in measurement approaches. However, this distinction may not be theoretically generalizable to any psychometric study on SR, where method variance may be better controlled by the use of a more comprehensive multivariate assessment battery.’

Ll. 49-51: reference missing

We added three exemplary references regarding brain-behaviour associations in the domain of cognition, personality, and psychiatric disorders in line 51.

l. 58: I would extent the statement on the aim/benefit of the current research: providing a well-founded measurement model is not only advantageous for statistical analysis but also for drawing valid conclusions based on neuropsychological research on the ABCD study.

We extended the sentence in line 55-59 and rephrased it slightly to enhance clarity and readability.

‘Since researchers analyzing data from these neuroimaging initiatives often have expertise in neuroimaging but less so in psychometrics, proposing well-established and robust psychometric models—along with implementation code—can enhance the statistical analysis of brain-behavior associations and support valid conclusions in neuropsychological research using the ABCD study.’

ll. 113: The usage of “content domain” is slightly misleading although it corresponds with the term content validity. DoG-measures and EF measures refer to slightly different aspects of self-regulation but also to different measurement approaches.

We appreciate this observation and have revised the text to explicitly address both content and methodological domains. Specifically, we have reworded the sentence in lines 114-115 to clarify this distinction:

‘Research on the convergent validity of SR measures reveals weak correlations between different content- and methodological-domains [24,26].’

Additionally, to ensure that the overlap between content and methodological domains is acknowledged, we have added “from different content domains” in lines 131–133:

‘In light of these considerations, combining task-based and survey-based SR measures from different content domains poses both opportunities and risks.’

ll. 119: with the explanation of the reviewers I better understand why they have chosen to write about content validity instead of construct validity. However, because including measurement approaches (partly) referring to different construct facets and testing whether the measurement model represents one general latent construct (what somehow does include the idea of convergent validity), at some point they should refer also to construct validity and not only to content validity.

We believe that the whole measurement approach is guided by considerations of convergent validity. Therefore, we have restructured the paragraph ‘The measurement of self-regulation’ in lines 112-135 to more consistently and explicitly address problems with content- and also convergent validity, e.g., in lines 114-115:

‘Research on the convergent validity of SR measures reveals weak correlations between different content- and methodological domains [24,26].’

Or 121-123:

‘However, the consistently low correlations between survey and behavioral SR indicators [24] point to issues with convergent validity.’.

Methods:

Table 1: I recommend to add one brief statement in the table notes explaining that parenthetical information in the firs column refer sth. referring to the ABCD dataset (which is, what I suppose)

To enhance clarity, we added a statement

‘Parenthetical dataset names in the first column indicate the ABCD dataset-derived short names of the data structure from which the data was retrieved.’

in line 296-297 in addition to the column title ‘Test (ABD dataset name)’.

Minor remark: “lavaan” in line 308 seems to be written in another font than the rest of the text.

We adapted the font to the rest of the text.

Discussion:

ll. 545-548: hot and cool system do not mainly refer to long vs. short-term orientation but to the extent to which affect/emotions are involved or rather cognitively focused – mainly referring to long-/short term may be misleading because from a conceptual view basic EF are often considered to be part of the cool system without being long-term oriented per se (but of course helpful to reach long-term goals).

Thank you for pointing out this potential confusion. In that sentence, our intention was not to highlight differences between the Hot and Cool factors—given that they are correlated—but rather to explain the distinction between them in terms of contextualized versus decontextualized control. This distinction was meant to explain the lack of correlation between Cool and EF. The original wording was, however, misleading, so we have rephrased the sentence to achieve more clarity – see lines 554-559:

‘This dataset-specific representation of the Cool factor in terms of broad, decontextualized and long-term control mechanisms, along with the more obvious issue of method-specificity, may also explain the lack of association between Cool SR and EF, despite their close theoretical link. SR captured by behavioral task performance, as measured with EF, is less closely related to delayed (as captured by the Cool factor) than to immediate SR, as they operate on different temporal scales of SR. ‘

7. PLOS authors have the option to publish the peer review history of their article (what does this mean?). If published, this will include your full peer review and any attached files.

Do you want your identity to be public for this peer review? For information about this choice, including consent withdrawal, please see our Privacy Policy.

Reviewer #1: Yes: Julia Grass

We have uploaded the files and PACE indicated that they comply with the PLOS requirements.

---

## [Editor Report · Decision Letter 2]

28 Mar 2025

The measurement of self-regulation in the Adolescent Brain Cognitive Development (ABCD) Study

PONE-D-24-12171R2

Dear Dr. Marek,

We’re pleased to inform you that your manuscript has been judged scientifically suitable for publication and will be formally accepted for publication once it meets all outstanding technical requirements.

Kind regards,

Iftikhar Ahmed Khan

Academic Editor

PLOS ONE
---

## [Editor Report · Acceptance letter]

PONE-D-24-12171R2

PLOS ONE

Dear Dr. Marek,

I'm pleased to inform you that your manuscript has been deemed suitable for publication in PLOS ONE. Congratulations! Your manuscript is now being handed over to our production team.

Kind regards,

on behalf of

Dr. Iftikhar Ahmed Khan

Academic Editor

PLOS ONE